# An Innovative Coded Language for Transferring Data via a Haptic Thermal Interface

**DOI:** 10.3390/bioengineering12020209

**Published:** 2025-02-19

**Authors:** Yosef Y. Shani, Simon Lineykin

**Affiliations:** Department of Mechanical Engineering & Mechatronics, Ariel University, Ariel 4077625, Israel; yosesh@gmail.com

**Keywords:** data transfer, haptic thermal interface, thermoelectric cooler, thermal cues, thermal patterns, thermal pulses, thermal icons, thermal communication

## Abstract

The objective of this research was to develop a coded language, similarly to Morse or Braille, via a haptic thermal interface. The method is based on the human thermal sense to receive and decode the messages, and is to be used as an alternative or complementary channel for various scenarios in which conventional channels are not applicable or not sufficient (e.g., communication with the handicapped or in noisy/silent environments). For the method to be effective, it must include a large variety of short recognizable cues. Hence, we designed twenty-two temporally short (<3 s) cues, each composed of a sequence of thermal pulses, meaning a combination of warm and/or cool pulses with several levels of intensity. The thermal cues were generated using specially designed equipment in a laboratory environment and displayed in random order to eleven independent participants. The participants identified all 22 cues with 95% accuracy, and 16 of them with 98.3% accuracy. These results reflect extraordinary reliability, indicating that this method can be used to create an effective innovative capability. It has many potential implications and is applicable immediately in the development of a new communication capability, either as a single-modality thermal interface, or combined with tactile sensing to form a full haptic multisensory interface. This report presents the testing and evaluating process of the proposed set of thermal cues and lays out directions for possible implementation and further investigations.

## 1. Introduction

As the modern environment becomes more and more technological, many activities involve data transferring via human–machine interfaces (HMIs), in which data is presented by a designated device and acquired by human senses. These activities can be divided into two general scenarios and hence two general purposes: virtual/augmented reality (VR/AR)—in these cases the presented stimuli simulate characteristics of the environment, causing the desired perception (e.g., in gaming, military uses, robotic or distant surgery, etc.) and communication, where the stimuli represent predetermined messages or information. The volume and versatility of the data are continuously expanding as technology advances, hence the requirements for available channels and methods. This study focuses on the communication application, specifically in the thermal medium.

The growing need for an assortment of new communication capabilities has motivated many research and development activities in the field of tactile interfaces [1,2,3,4,5,6]. However, despite the increasing interest, the thermal haptic field has yet to receive sufficient attention.

The thermal haptic sensory modality offers a novel dimension for transferring information, provided the thermal cues are designed in accordance with the human sensory system properties. The thermal sense naturally serves as a data transferring medium in our everyday routine. It takes part in perceiving the environment, assessing the temperatures of surrounding objects, and recognizing the material they are made of. Intensive research has been addressed over decades to understand the physiology and psychophysical processes associated with the human thermal sense, showing remarkably high sensitivity and resolution. Humans are surprisingly susceptible to changes in skin temperature, especially for cooling [7]. Neurophysiological studies of thermo-receptive fibers showed that when our skin is stimulated by successive pairs of cooling pulses, the fibers affiliated with sensing coolness can resolve extremely small differences of 0.02–0.07 °C between them and differences of 0.03–0.09 °C in warmth pulses by ‘warmth fibers’. As to absolute detection, we can detect thermal stimuli when skin temperature rises by 0.2 °C or descends by 0.11 °C (at rates above 0.1 °C/s) [8,9,10,11]. These values are highly dependent on various objective and subjective parameters, some of which relate to the participants, such as the current state of mind, skin temperature baseline, variance between people, etc., and others relate to experimental limitations. However, these values provide a sense of the natural human capabilities we wish to harness for the development of a communication medium.

The importance of developing a method to effectively use the human thermal sense as a data transfer medium derives from the many potential implementations it is expected to have as an alternative or complementary channel for various scenarios in which conventional channels such as vision, hearing, and tactile sensing are not applicable or not sufficient (e.g., enhancing communication capability for the deafblind [12], transferring messages in silent/discrete scenarios [5], or alternatively, in noisy/bumpy environments [13]).

The leading bioinspired approach for VR/AR applications is to combine senses to increase immersion and enhance the virtual experience. This approach was proven efficient for haptic communication as well, for applications requiring large haptic cue sets. A good example to demonstrate the implementation of this approach is the multisensory tactile prototype mentioned above [1,2,6].

A survey of the reported applied research on thermal communication and other applications of data transferring via a thermal haptic interface was recently conducted [14]. These reports showed that combining thermal feedback with other types of feedback can improve performance in various applications. Adding thermal information regarding objects being virtually touched can increase the feeling of immersion, and simple thermal messages indicating locations or emotions can be of use. The authors concluded that the potential interest of thermal devices lies in the areas of both gaming and communication, and that there is much room for further research.

The general goal of this research was to develop a method to effectively use human thermal sense as a data transfer medium. The proposed method is designated to be implemented as a single-modality capability or alternatively assimilated into a multisensory capability. For the capability to be effective and robust it must include a large variety of short recognizable cues. These three parameters—variety, duration, and recognizability—are the leading characteristics in the design and evaluation of the thermal cues.

Developing a capability of generic data conveyance via the thermal haptic sensation means using a thermal display to present encoded abstract information to be sensed and decoded by touch. A thermal display can take various shapes and forms: portable and accessible devices such as a mobile phone (stimulated by Peltier modules) [14,15], a wearable item such as a watch or bracelet [16], a clothing item fabricated from a smart textile with active thermal actuators and embedded devices [17,18,19], or stationary elements typically in contact with the user, for instance, a vehicle or theater seat, etc. The most commonly used technology in research is Peltier thermoelectric coolers/heaters (TEC). They are characterized by a fast and reliable response to the electric driving current stimulus, forming a corresponding thermal signal.

The concept of cues composed of sequences of thermal pulses was chosen, several cues were designed, and a preliminary feasibility test was conducted [20]. The cues must be knowledgeably designed to guarantee perceptual distinction between them with high reliability. The initial pulses and patterns were designed with the aid of an original modeling system [21]. The designs were then improved via a repetitive trial-and-error process during which the various parameters were adjusted to find the optimal tradeoff for the minimum cue duration and maximum recognizability. The varying parameters are the rate of skin temperature change, or equivalently, the amplitude of the stimulating current, the duration of each pulse, and the time interval between pulses in each sequence composing a cue. The degree of pattern recognition derives from the ability to notice every pulse in the sequence and refer to each pulse individually and the ability to distinguish between varying intensities. Due to practical reasons, the tests at this stage involved only cooling pulses.

The final patterns evaluated in the feasibility test consisted of two or three cooling pulses with a total duration of up to 2.75 s. The cues were found to have a low impact on skin temperature (up to 2 °C), which is a significant advantage for future implementation; nevertheless, they were strongly noticed and clearly recognized. The tests were limited in the manner that they involved only a few participants and were based purely on cooling pulses but were sufficient to provide the necessary data to proceed to the next phase of the research.

Twenty-two different cues were then designed based on the findings and insights from the feasibility test, and a reliability test was conducted. The recognizability and robustness of the cues were evaluated by a group of volunteers. This paper presents the test results showing the cues’ reliability and hence suitability to serve as a generic data transferring method.

## 2. Methodology

An absolute identification experiment was conducted in which participants were asked to identify which one of the twenty-two thermal patterns was displayed. The cues were displayed single blindly in an arbitrary order, and the whole series was displayed twice for each participant. The random order was generated in MS Excel using the RAND function. The participants were pre-acquainted with the cues; familiarized with the thermal sensation evoked by each cue, and experienced matching a thermal sensation to the corresponding serial number using a user-friendly cue-chart (see Section 2.3). To simplify the identification process and minimize technical errors, the serial number was accompanied by a matching graphical description associated with the thermal sensation and a phonetic code logically indicating the specific sequence. The responses were acquired and statistically analyzed using a confusion matrix.

### 2.1. Participants

Eleven paid volunteers (6 female and 5 male) participated in the experiment (average age 28.4 years, range 18–52 years). All participants reported no known abnormalities of their tactile and thermal sensory systems. One tended to relatively cold palms, but this slight thermal sensitivity was overcome using external warming (and cooling) devices to guarantee identical initial skin temperatures for all participants, at all iterations throughout the entire experiment. All subjects gave informed consent to take part in the research, and the experiment was conducted in accordance with the approval granted by the Ariel University Institutional Ethical Committee for Nonclinical Research on Humans.

### 2.2. Apparatus and Test Layout

A Laird Thermal Systems model UT-15-200-F2-4040-TA-WG Peltier thermoelectric cooler (TEC) was used to create the thermal display. One side of the Peltier element served as the display, and the other side was mounted on a thermostatic hot plate TE Technology CP-200HT using thermal grease to guarantee full thermal contact between them. The stabilized temperature was maintained by a TC-720 controller using a thermistor to monitor the interface temperature. Another identical thermistor was attached to the hot plate surface at some distance from the TEC, indicating the controlled temperature. The hot plate’s controlled temperature was set to 33 °C, so that the thermal display temperature at a steady state with the ambient air of 21 °C was 32 °C.

The NI myDAQ Student Data Acquisition Device was used as an electrical signal generator. Pre-designed voltage stimuli were formed by the myDAQ and delivered to the thermal display via a custom-made current driver. The driver was designed for constant input in the range of −15 A to +15 A or peaks up to ±20 A and to amplify the signal with a sensitivity (Iout [A]/Vin [V]) of 3 [A/V] (see Figure 1). An Agilent 34792a Data Acquisition System with Agilent commercial software was used to collect the data and process it for analysis. The temperature change throughout the experiment was monitored at various positions using MP 3176 thermistors (5 kΩ, 0.9 mm diameter). The thermistors were mounted on the thermal display with layers of Aerogel thermal insulator to thermally isolate the zones of interest (see Figure 2). The general view of the laboratory layout is shown in Figure 3.

### 2.3. Thermal Cues—Designing Considerations

A set of twenty-two different sequences of thermal pulses was tested in this study. The cues were designed in light of the results and insights from the preliminary feasibility test [20], which served as a guide in determining the feasible sequences and the intensities and time durations of the pulses and of the gaps between pulses for each specific combination. The resultant specifications were fine-tuned using an iterative trial-and-error process that included empirically verifying the cues’ uniqueness. As a result of this process, 22 cues were selected out of all the possible combinations. They included three categories of sequences: solely warmth pulses, solely coolness pulses, and alternating warmth and coolness pulses. The first two categories, i.e., the thermally homogeneous cues, were constructed of either one, two, or three pulses, at one or two levels of intensity. Serving the goal of minimal cue durations, all pulses were designed with a unified electrical current of 15A, and the intensity was determined by varying the duration of the stimulus. The actual stimulus durations were not limited to two fixed values but were rather individually determined for each pulse so that the desired thermal relative sensation was reached, forming the desired full cue perception. The warmth and coolness alternating cues included only one level of intensity. As in the other two categories, the actual stimulus durations were not unified but were rather individually determined so that the pattern would be clearly identified. In fact, the pulse durations in the whole set of cues varied from 30 ms to 450 ms. The total cue duration was under 3 sec, with an average of 2.03 s. The cues were presented in a chart specially designed as an aid for the identification process, with a graphic description and a coding system that includes symbols, serial numbers, and colors—see Figure 4. The iconic graphic expression intuitively corresponds to the thermal sensation the cue causes rather than to the electrical stimuli or the skin temperature change. An exception was made regarding the number of intensities per sequence, as it was found that in the case of unidirectional intensification, i.e., either increasing or decreasing intensities (cues #6, 7, 13, and 14 in Figure 4), it is feasible and even preferable to use three intensities.

### 2.4. Experimental Protocol

Preliminary familiarization: The participants were first acquainted with the set of cues. After a brief introduction and a hands-on experience, they were expected to comprehend the coding method and be familiarized with the thermal sensations. The familiarization routine was unified for all participants. It included the exposure to a consistent sequence of cues: three coolness cues—#8, 9, and 10; three warmth cues—#1, 2, and 3; and finally, two thermally alternating cues—#16 and 21. These cues were chosen to provide a sense of the basic features that include coolness pulses, warmth pulses, varying intensities, and alternating direction. Following this generic routine, the participants had a 15 min period to freely experience various cues by choice.

Safety measures were embedded in the experiment design and were explained to the participants. The presented cues were all chosen out of a pre-programmed arsenal, and all hardware preset conditions were determined and empirically tested by the experimenter prior to every session. Thermal display temperature was always kept within the comfort range of 26–39 °C. In addition, the temperature was continuously monitored and observed by the experimenter. Nevertheless, participants could simply withdraw their hand at any given time if they felt uncomfortable (due to thermal illusions, etc.), thereby eliminating any risk of pain or harm.

Thermal baseline: Prior to every iteration, the skin temperature of the hand at testing was required to be within the range of 32 ± 0.3 °C. This initial condition was reached by the participant with the aid of external warming/cooling devices, after the skin temperature was manipulated during the previous cue. The participant was responsible as well for retaining or regaining a neutral thermal feeling in the hand (the entire hand, not only the area to be in contact with the display, because otherwise a distorted thermal perception would be formed when exposed to the cue). In addition, the skin temperature at the contact zone was verified by the experimenter as contact was made with the thermistor. If a drop or a jump was observed in the monitored temperature that is displayed on the Agilent front panel, then the participant was instructed to improve skin temperature and resume contact.

Test procedure (see flowchart in Figure 5): The participant rested a given hand on the display, carefully positioning the thenar eminence at the base of the thumb in a natural motion. The guidance was to rest the hand but not press (the actual pressing force was not monitored), while ensuring full skin contact with the display and coverage of the entire area. By this act, the monitoring thermistors mounted on the display (see Figure 3) were inevitably located at their respective functional positions. This position was maintained steadily throughout the present iteration. After a non-consistent and non-predictable period of several seconds, the thermal cue was displayed. The presented cue was randomly chosen (using the convenient RAND function in Excel that generates a random variable with normal distribution). The participant then was required to identify the cue and report the corresponding serial number using the cue chart as an aid. At the participant’s request, the cue was displayed again, after verifying a return to the thermal baseline. The number of repetitions was not limited but rarely exceeded 4. The initial identification and the final one were recorded. This procedure was repeated for each participant until all 22 cues were displayed and recognized twice.

### 2.5. Data Analysis

The main data in this study included the initial and final identifications the participants made for each cue based on their subjective thermal perception. In addition, more data were gathered during the experiment for post-analysis and research purposes, including the stimulus specification in terms of the input current and the consequent voltage on the Peltier TEC and the temperature changes monitored at the various locations during exposure to the stimuli.

To evaluate the performance of the suggested set of thermal cues, the cue-recognition accuracy was determined using a confusion matrix (22 × 22). The matrix arranges data gathered from all the participants and presents the identifications vs. the actual cues displayed. The value in each cell represents the ratio of identifications to the total repetitions per cue (which is 22, two for each of the eleven participants). This layout depicts the level of success in identifying each cue, in percentage units, as the main diagonal shows the accurate identifications, and the rest presents the incorrect ones. Two matrixes were formed, one for the initial identifications (i.e., after a single exposure) and one for the final identifications (i.e., after several exposures required by the participant to reach a high level of certainty). This tool allows analysis between variables at two levels of resolution: comparing the performance of the three groups of cues—warmth, coolness, and alternating—and comparing the different cues individually. An additional measure for the quality of performance, other than the recognition accuracy, can be the recognition clarity based on the number of repetitions required per cue or per group of cues.

The analysis between people included examining the impact some characterizing parameters had on recognition quality. The collected parameters that are relevant to haptic thermal sensitivity are age, gender, BMI, and subjective reported thermal sensitivity.

An additional analysis for research purposes was aimed at better understanding human thermal behavior and was not directly related to the evaluation of our set of cues. The physical skin response to the various thermal cues, indicated by the skin temperature change, was monitored throughout the experiment. The correlation between the physical response and the psychophysical response forming the thermal perception was evaluated by analyzing the skin temperature change vs. the recognition precision. This analysis included a comparison between participants and between cues.

## 3. Results

The experiment included 11 participants that were exposed twice to a set of 22 different thermal cues in a random order and were required to identify each one of them by touch. The data received from these tests included 44 initial and 44 final cue identifications per participant (with an average overlap of 15 due to cases where a single exposure was sufficient for decisive identification and no further repetitions were required; hence, the initial identification was the final one as well), along with the temperature change monitored at the interface throughout the process.

### 3.1. Cue Identification

The identification results were arranged in confusion matrixes—see Figure 6. The results show an identification accuracy of 84.3% for the initial identification and 95.0% for the final identification. In total, 10 cues out of 22 were perfectly identified by all participants with zero mistakes. When filtered by removing cues that scored less than a 95% accuracy (a total of six cues were removed: three warmth—#2, 3, and 5; one coolness—#10; and two alternating—#17 and 18), a set of 16 cues remained, showing a final identification accuracy of 98.3%. See Table 1 for a breakdown of the accuracy performance according to the different cue categories.

The number of times the participants requested to repeat a given cue until reaching a decisive identification was recorded. In many cases, the final identification did not change (no significant regularity was found, and the uncertainty seems to be personality related); in many others, it did. Hence, the supremacy evidenced in the final identifications vs. the initial identifications. However, when comparing cues, this parameter appears to represent a measure of certainty the participants had in their identification, indicating cue robustness. Figure 7 shows that the warmth cues required an average of 2.2 repetitions, the coolness cues required 1.7, and the alternating cues only 1.4. Within the three categories, Figure 7 reveals the cues that were relatively more inconclusive—cues #2, 4, and 7 of the warmth cues, #11 and 13 of the coolness cues, and #15 and 18 of the alternating cues. It should be noted that in many cases of uncertainty throughout the experiment, a short break ‘reset the sensing system’, and the fresh exposure enabled identification easily and correctly and with high certainty.

The correlation between some parameters characterizing the participants and their individual performance is shown in Figure 8. Although it appears that age and BMI are correlated with a reduction in identification accuracy, these correlations are statistically insignificant (*p* >> 0.05).

### 3.2. Skin Response

The temperature change that occurred in the skin in response to the thermal stimuli is indicated by the temperature monitored at the skin–display interface. Figure 9 presents an example of data from several representative cues. The graphs include the interface temperature change plotted in comparison to the thermal cue itself as monitored on the display (see Figure 2—thermistors #2 and 1, respectively). Each curve includes a flat phase that represents the last several seconds of skin–display contact awaiting the thermal cue and three waves corresponding to the three pulses constructing the cue. The initial phase of each wave represents the jump or drop in skin temperature following the corresponding pulse, and the rest of the wave represents the recovery process following the termination of the pulse. The recovery to baseline temperature after the last pulse is only partial, as the final skin temperature is dependent on the nature of the last pulse. Natural thermal recovery driven by physiological thermoregulation is a prolonged process, even though exhilarated by contact with the thermal display that is forced toward the baseline. The thermal sensation caused by a given pulse, as illustrated by the graphic icon, is basically proportional to the extent of the jump/drop relative to the current starting point, regardless of the actual measured temperature. The total temperature change per cue varied from cue to cue but was (at the skin–display interface) within the approximate following borders: +4 °C for warmth cues, −2.5 °C for coolness cues, and ±2 °C for alternating cues.

The rate of temperature change (ROC), at the skin–display interface, was calculated and plotted in Figure 10 (the same three representative cues previously demonstrated). ROC is the derivative of the temperature change curve to time. Naturally, the initial phase of the ROC curve is flat at the value of zero, indicating a thermal steady state.

The skin response was substantially uniform for all participants. Figure 11 presents an example of data from the same three cues presented in Figure 9, showing the skin temperature change for five arbitrary participants (more than five would make it difficult to visually notice details). The uniformity is not absolute, as differences of up to approximately 10% can be seen in the temperature change. The majority of these slight variances in the actual measured temperatures are due to the different starting points that are visible along the entire first phase prior to the stimuli. Nevertheless, synchronizing the temperatures by shifting the readings up or down according to the actual starting temperature to meet at a precise 32 °C reveals the remaining deviations between participants but also within participants between two appearances of the same cue. Figure 12 demonstrates an extreme case in which the uniformity between participants is greater than within participants.

## 4. Discussion

The concept of sequences of short thermal pulses implements advantages observed in other methods and approaches and compensates for inherent relative disadvantages of the thermal medium. The recognizability of different sequences is based on the proven human ability to distinguish well between warm and cool stimuli, between single and multiple stimuli, and between successive pulses with different intensities. This research aimed to acquire the scientific knowledge required to design robust patterns that can guarantee reliable identification and to design and build the means to create and evaluate them.

The results from the experiment conducted in this study show that we have managed to develop a basic capability of generic data conveying based on thermal sensation. A full applicable capability requires further development that includes three major aspects, with mutual impact: assimilating improvements in the cue design, determining whether to combine the thermal cues with other haptic senses or apply them as a sole medium, and developing appropriate hardware. The development of an operational applicable solution will also include intensive testing with a larger group of participants to validate the results with higher statistical power than the current work, which used only eleven participants. However, the extremely high accuracy shown in identifying the set of cues, by untrained and inexperienced users, clearly indicates the extent of the breakthrough.

The following features of the pulse-based cues were learned from the preliminary feasibility test previously conducted [20] and completed during the iterative designing process forming the final set of cues for the current experiment:

Current–duration tradeoff—the stimulus current amplitude is equivalent to the heat flow rate and dictates the change in skin temperature at a steady state. But for temporally short pulses in which the process is terminated earlier, the duration actually impacts the ROC. Therefore, the pulse intensity can be set by the current–duration tradeoff.The pulse position in the sequence impacts the sensation in two counteracting ways: (1) the perceived intensity of a given pulse increases as its position in the sequence is progressed; (2) for cues that combine both warmth and coolness pulses, the perceived intensity of the alternating direction cue is degraded.Intervals between pulses—the optimal frequency for minimal cue duration and maximal recognizability is roughly a pulse per second. Fine-tuning is required per sequence, corresponding to the pulse intensities. Due to different temporal responses for warmth vs. coolness stimuli, when alternating from warmth to coolness, the interval should be extended (by approximately 300 ms), and vice versa, in order to maintain the perceived optimal time intervals.

Between the three cue categories, it seems the warmth–coolness alternating cues were the most recognizable. This is apparent from the identifying accuracy (see Table 1), as well as the extent of certainty the participants felt with their identification (see Figure 7). This finding is coherent with [22,23], who showed that the direction of change in temperature (warming or cooling) is the most salient parameter in thermal cues. Following, with a very slight gap but consistent in both parameters, were the coolness cues. The warmth cues were substantially the weakest category according to the results, and in addition, they required greater intensities—extended electric stimuli and consequently greater temperature changes.

Reviewing the incorrect identifications presented in the confusion matrix (Figure 6b) reveals that the confusions were usually not mutual (e.g., cue #10 was mistaken for cue #11 but cue #11 was identified accurately, cue #12 was mistaken for cue #14 but #14 was identified accurately, etc.), except for the warmth cues #2, 3, and 4, where cue #2 was mistaken for cue #3 and cue #3 was mistaken for cues #2 or 4 and cue #4 was mistaken for cue #3. Participants’ remarks in regard to these confusions included ‘false negative’ perceptions, e.g., the two warmth pulses in cue #18 were interpreted as one and hence identified as cue #15. On the other hand, ‘false positive’ perceptions, e.g., cue #3 was identified as cue #4, occurred as the intensity of the second pulse caused a residual sensation of another pulse. In addition, some specific cues appeared to be preferred choices (e.g., for two out of three misidentifications in the coolness category, cue #11 was chosen; for three out of four misidentifications in the alternating category, cue #15 was chosen). These findings indicate that with some training and experience, the cues should be expected to build up robustness and reliability, that is, improve identification certainty and accuracy as well as reduce the necessity of meticulous initial conditions.

The approach taken in this research for creating controlled thermal stimuli was to use a voltage-controlled current driver that delivered a stabilized current to the thermal display. The voltage was set according to the cue design, and the current was determined proportionally, regardless of the load applied by the skin in contact with the display. Since the heat flux from the display into the skin is practically proportional to the input current (assuming it is unidimensional, neglecting losses due to heat distribution through the skin surface, as well as the losses due to electrical and thermal resistance of the TEC itself that develop with time), the research assumption was that creating a controlled current means creating a controlled heat flow through the skin. The results showed a uniform skin response for all participants (see Figure 11) regardless of their cutaneous thermal characteristics and regardless of the contact thermal resistance, confirming this research hypothesis.

Furthermore, the uniform physical response to the different thermal cues (see Figure 11) indicates that a simulator [21] can potentially play an important role in future cue design. Its reliability can be extremely high since the response is indifferent to the inevitable natural variance in cutaneous thermal characteristics among people, leaving only the correlation between physical response and the consequential thermal perception to be resolved.

The tested cues had a relatively small impact on the skin surface temperature (up to −2.5 °C for coolness cues and up to +4 °C for warmth cues). Nevertheless, it is safe to assume that the thermal signature can be even reduced by lowering pulse intensities while maintaining identification accuracy. Implementing AI capabilities where applicable will enable minimizing intensities and durations by optimizing the stimuli to the individual thermal perception.

A set of twenty-two generic cues is sufficient to create a full capability of data conveyance (the ancient Biblical Hebrew language as well as the modern Hebrew contain twenty-two letters). However, these same cues can be used as individual messages as well.

## 5. Conclusions

A novel capability of conveying generic data via a thermal haptic interface was developed. The full capability has yet to be fully explored, but the basic method has been proven effective and applicable. This method can be implemented as a single-modality capability or alternatively assimilated into a multisensory capability. The developed method is composed of cues that contain encoded information in an abstract form, to be sensed and decoded by touch. The cues answer to all three parameters that define the effectiveness of this sort of capability: variety, duration, and recognizability. The empirical results show extremely high accuracy in identifying 22 different cues (95%), 16 of which were close to absolute identification (98.3%).

The significance of this achievement is in two major aspects. The first aspect is the need—there is an undisputed growing need for new communication capabilities for various scenarios and implications. This need has motivated research and development activities in the field of tactile interfaces. The second is the challenge. Such a capability must overcome many existing physiological, physical, and technological obstacles: the limited number of sensations evoked by changes in skin temperature, the multiplicity of parameters that influence human thermal sensitivity, poor spatial sensitivity, slow response due to heat transfer processes, thermal adaptation, creating and controlling the desired thermal stimuli, monitoring the skin response during the research without influencing the heat transfer process itself, etc.

Implementation of the developed method can be either as a stand-alone capability or combined with tactile interfaces [1,2]. Forming a full haptic multisensory interface can potentially leverage the relative advantages of each sense, enhancing noticeability and recognizability.

Furthermore, to enrich the variety of thermal cues and expand the potential domains of applicability, future research shall focus on two additional efforts:

Better utilizing the space dimension, i.e., enlarging the contact area with thermal displays thereby improving sensibility, and, more intere stingly, varying stimulation locations using wearable items (such as a watch, a bracelet, a ring, or an article of clothing formed of a smart textile). Encouraged by recent reports showing a certain level of spatial sensitivity [16,24,25,26], accounting for space–time interactions that may influence thermal perception [27] and for stimulus localizing limitations, e.g., when different fingers of the same hand are exposed to different stimuli vs. fingers of different hands [28].Evaluating new approaches for advanced thermal cues of different types, e.g., continuous periodical fluctuation of pulses with varying encoded frequencies, sequences of pulses at adjacent locations creating a perception of motion/direction, etc.

## Figures and Tables

**Figure 1 bioengineering-12-00209-f001:**
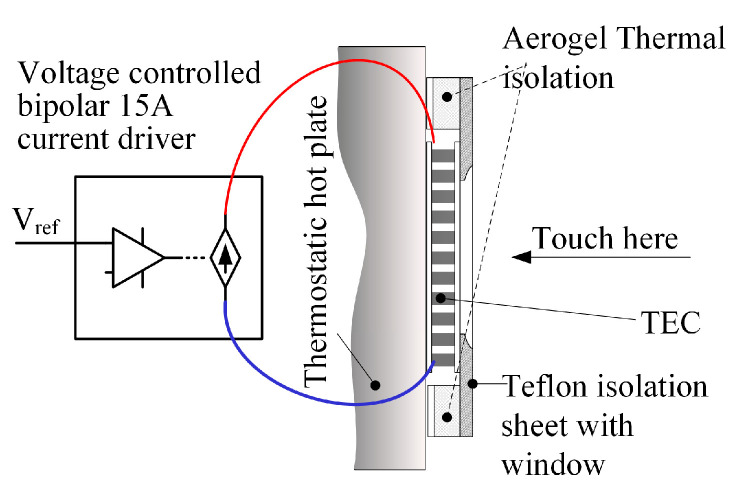
The experimental setup scheme includes the thermoelectric cooler (TEC), thermostatic plate, thermal insulation, and current driver.

**Figure 2 bioengineering-12-00209-f002:**
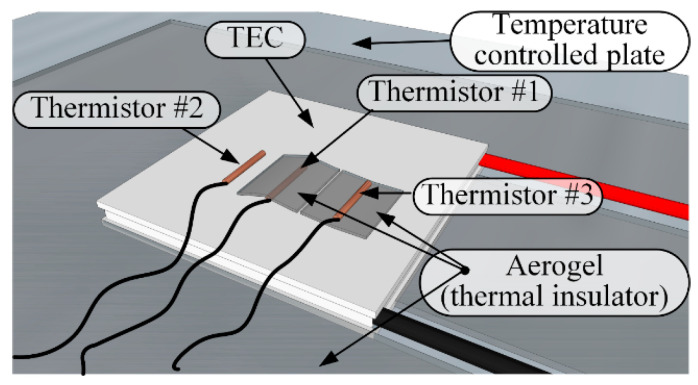
Scheme of the monitoring thermistor layout on the experimental thermal display. Thermistor **#**1—monitoring the temperature change of the thermal display when thermally isolated from the skin using a layer of Aerogel. Thermistor **#**2—monitoring display–skin interface temperature. Thermistor **#**3—monitoring skin temperature at the contact zone by isolating it from the display.

**Figure 3 bioengineering-12-00209-f003:**
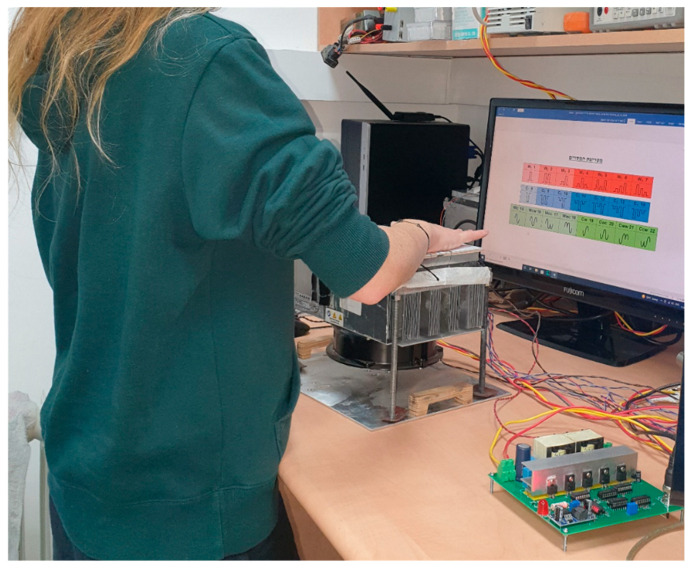
Laboratory layout—the picture shows the thermal display placed on the stabilized hot plate and covered by the participant’s right hand and the current driver that receives the signal from the voltage generator and transforms it to the input current for the display TEC.

**Figure 4 bioengineering-12-00209-f004:**
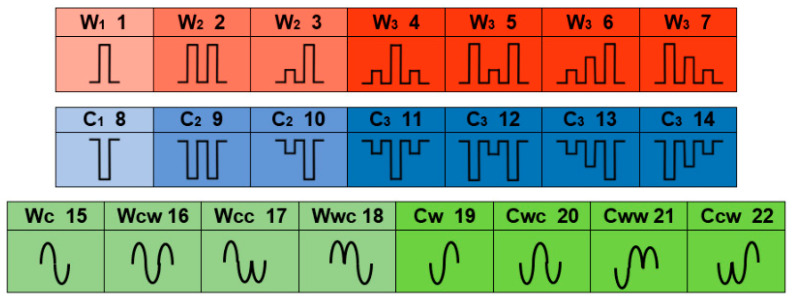
Cue chart—22 cues, marked by serial numbers from 1 to 22 (appears on the right top corner of each cue cell). Seven warmth cues, colored red (gradually deeper corresponding to the increasing number of pulses), coded by a ‘W’ for warmth, with a subscript index number of 1–3 representing the number of pulses forming the cue, and graphically presented as a sequence of pulses directed upwards (indicating warmth, as the temperature increases) and the relative pulse intensity. Seven coolness cues, colored blue, coded by a ‘C’ with a 1–3 index, icon directed down indicating coolness pulses, and eight alternating cues, coded by a sequence of the letters ’W’ and ‘C’ corresponding to the cue pattern, colored green (two different greens indicating the nature of the first pulse), and a graphic icon.

**Figure 5 bioengineering-12-00209-f005:**
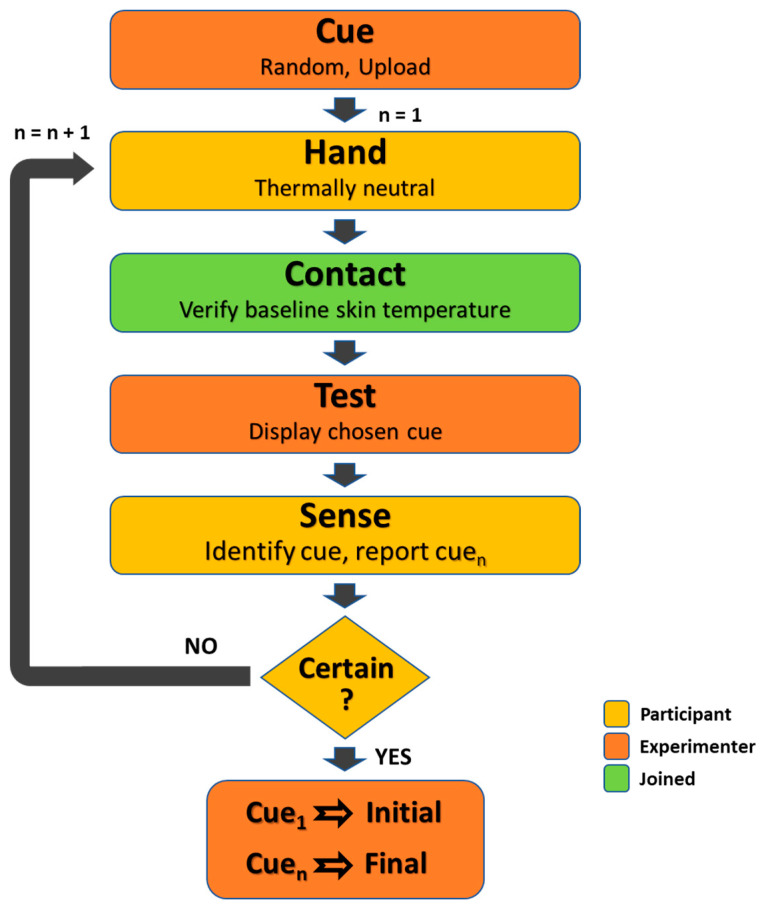
Flowchart illustrating the test procedure for a given cue—starting with randomly determining the cue and ending with documenting the participant’s initial and final identifications. In the case that a single exposure was sufficient, Cue_n_ = Cue_1_.

**Figure 6 bioengineering-12-00209-f006:**
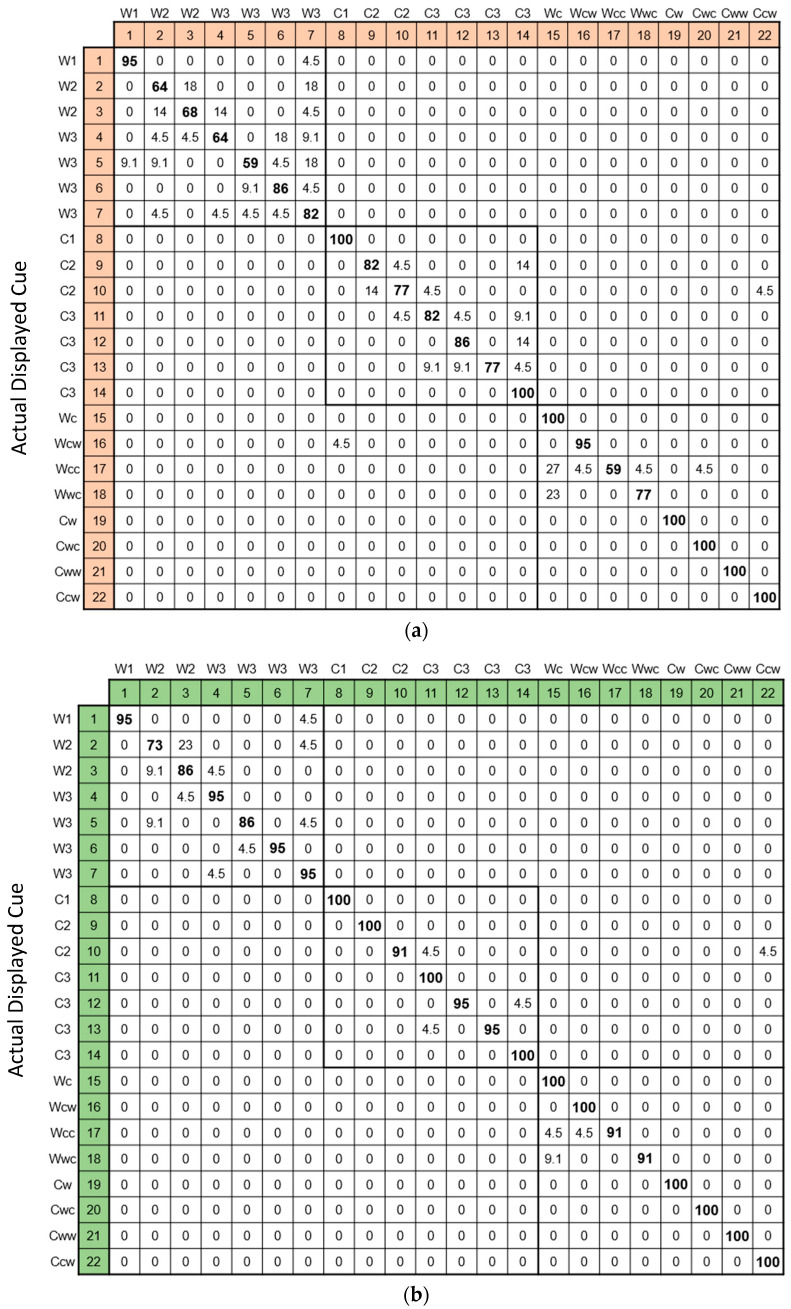
Confusion matrix—mapping the individual subjective identifications vs. the actual cues displayed, showing the collective level of identification accuracy per thermal cue. Two matrixes are depicted: for the initial identification after a single exposure to the cue (**a**) and for the final identification with high certainty (**b**).

**Figure 7 bioengineering-12-00209-f007:**
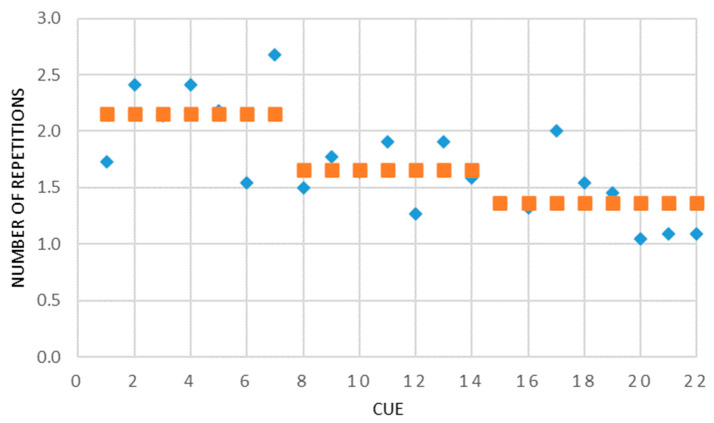
The number of repetitions as a measure of identification certainty—the average number of exposures per cue required by the participants to identify it with high certainty (blue) plotted together with the category averages (brown), showing a very definite tendency to have more difficulty in identifying the warmth cues, significantly less for the coolness cues, and even less for the alternating cues.

**Figure 8 bioengineering-12-00209-f008:**
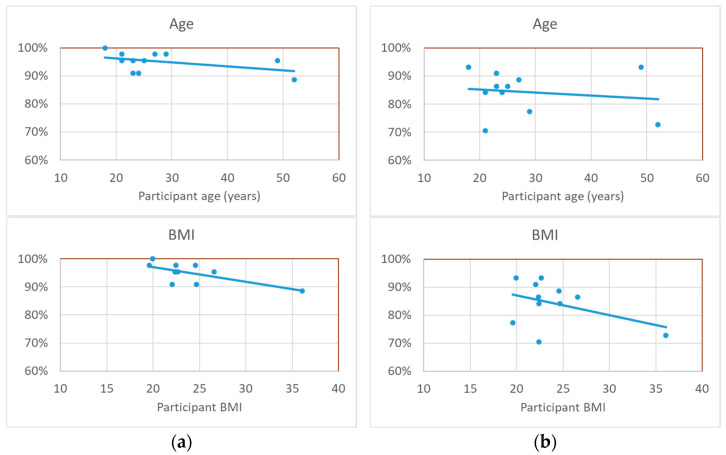
Individual identification accuracies of the participants vs. their age and vs. their BMI showing insignificant correlations: for the final identifications (**a**) as well as for the initial identifications (**b**).

**Figure 9 bioengineering-12-00209-f009:**
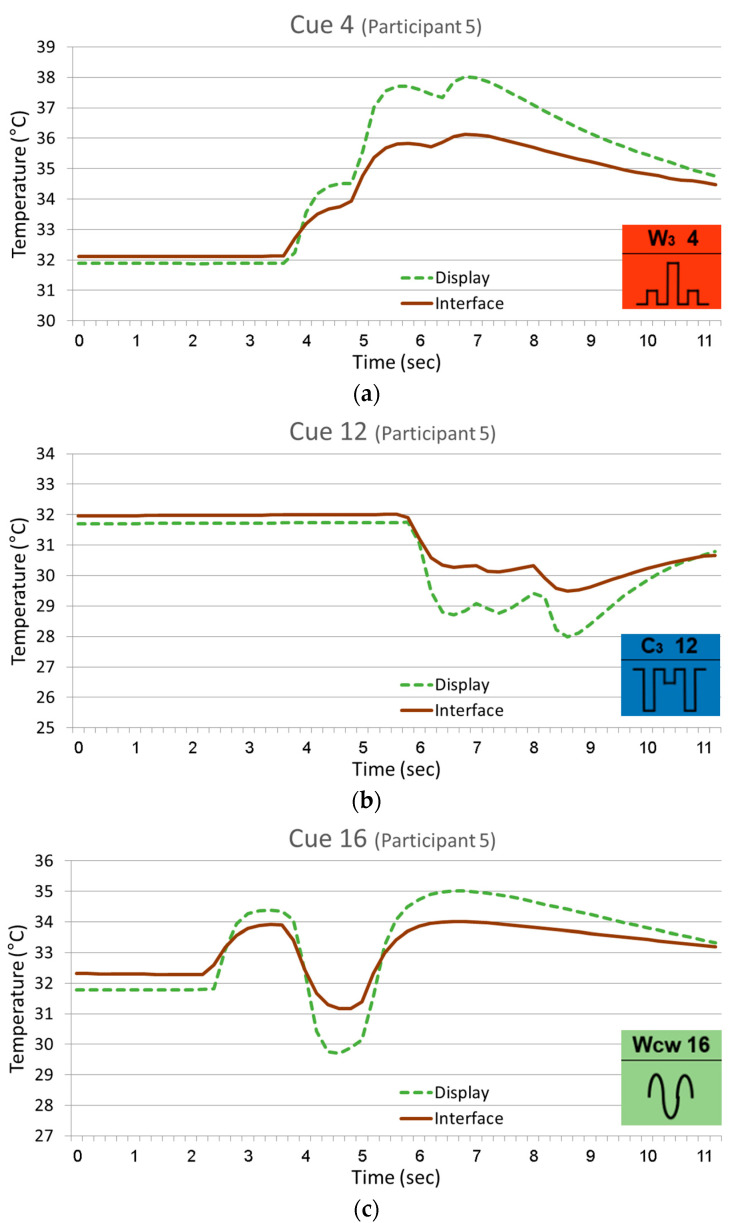
Skin response to thermal cues—the temperature changes monitored on the thermal display and at the skin–display interface are demonstrated with data from arbitrarily chosen participant #5 for three arbitrary cues representing the three categories—warmth cue #4 (**a**), coolness cue #12 (**b**), and alternating cue #16 (**c**). The corresponding graphic icon, phonetic code, and serial number from Figure 4 appear in the right-hand bottom corner.

**Figure 10 bioengineering-12-00209-f010:**
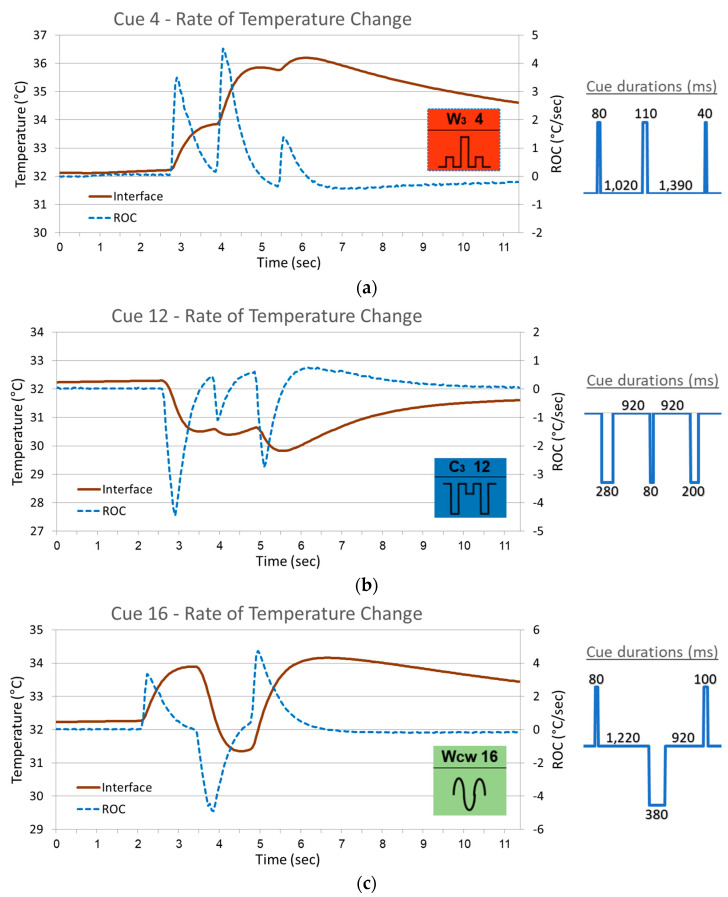
Skin response to thermal cues in terms of nominal temperature and ROC—the temperature change monitored at the skin–display interface and the rate of temperature change (ROC) are presented for the same three arbitrary cues presented in Figure 9, representing the three categories—warmth cue #4 (**a**), coolness cue #12 (**b**), and alternating cue #16 (**c**). Alongside the skin-response graphs are depicted the corresponding cue designs with the detailed time duration of each pulse and each interval.

**Figure 11 bioengineering-12-00209-f011:**
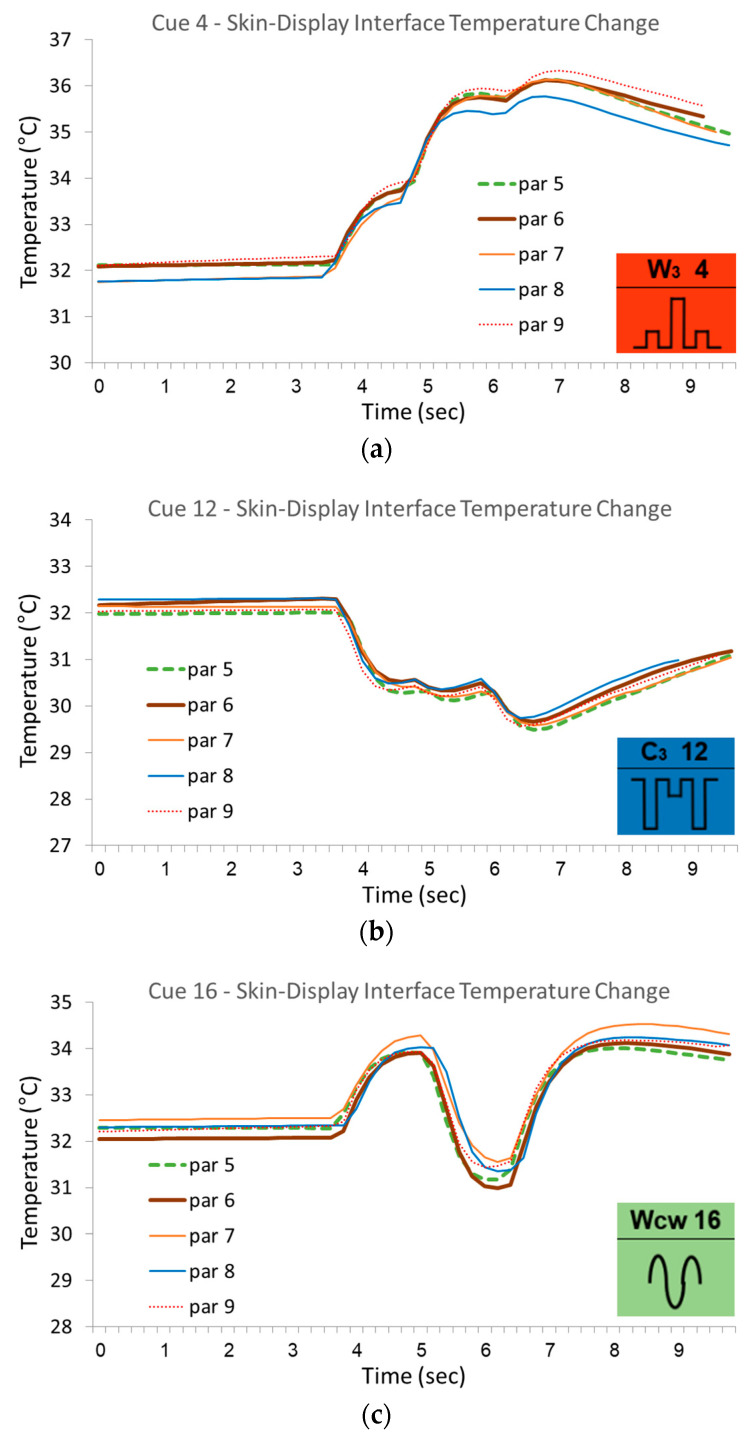
Comparison of skin responses for different participants - the temperature change monitored at the skin–display interface is demonstrated with data from five arbitrary participants for the same three arbitrary cues presented in Figure 9, representing the three categories—warmth cue #4 (**a**), coolness cue #12 (**b**), and alternating cue #16 (**c**). The data were temporally synchronized only by setting a common initial onset point, thereby showing the uniformity of the response.

**Figure 12 bioengineering-12-00209-f012:**
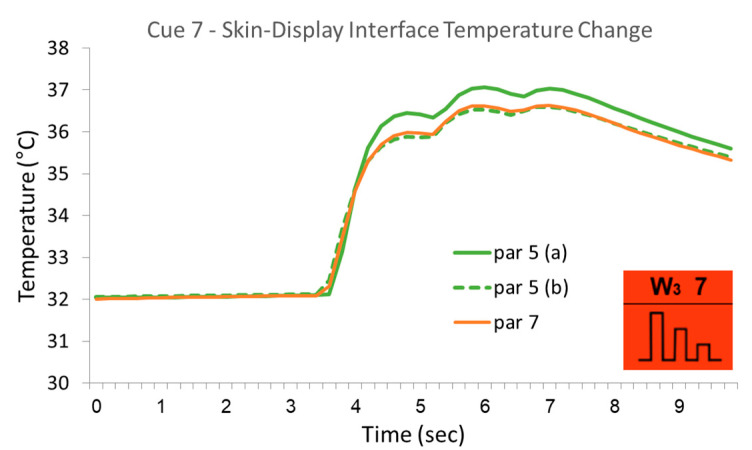
Variances in skin responses to a given cue from the same participant—the temperature changes monitored at the skin–display interface from the two cases where participant #5 was exposed to cue #7 are presented in comparison to the skin response of participant #7, showing extreme uniformity between participants versus a relatively rare difference between results within the same participant.

**Table 1 bioengineering-12-00209-t001:** Identification accuracy—breakdown of the different categories of cues.

Cue Category	Identification Accuracy		Final Accuracy for Filtered Set
	Initial	Final	
Total—set of 22 cues	84.3%	**95.0%**	**98.3%** (16 cues)
Warmth—7 cues	74.0%	**89.6%**	**95.5%** (3 cues)
Coolness—7 cues	86.4%	**97.4%**	**98.5%** (6 cues)
Alternating—8 cues	91.5%	**97.7%**	**100%** (7 cues)

## Data Availability

The data presented in this study are available on request from the corresponding author.

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
