# Peer review of "An Innovative Coded Language for Transferring Data via a Haptic Thermal Interface"

_bioengineering, 2025, doi:10.3390/bioengineering12020209_

Round 1

Reviewer 1 Report

Comments and Suggestions for Authors

The article need further clarification at certain points to consider to next level

1) How does a coded thermal language compare to traditional tactile or vibrational communication methods?

2) How do variations in skin sensitivity impact the effectiveness of thermal cues?

3) How might this technology be integrated into assistive devices for individuals with sensory impairments?

4) What advancements are needed to enhance the precision and responsiveness of thermal haptic interfaces?

Reviewer 2 Report

Comments and Suggestions for Authors

I suggest the manuscript to have some minor changes. The design of the experiment looks good, and the manuscript is well written. However, some parts need more explanation and improvement to make the work and its presentation stronger.

Clarification of Experimental Procedures: The experiment says cues are shown in "arbitrary order," but it would be better to explain more clearly how the order was randomized (for example, using random number generation or counterbalancing). This way, the readers will understand if there were any controls for possible ordering effects. It would also be useful to say if participants had practice trials before the experiment started. This could affect their ability to reliably recognize thermal patterns.

Statistical Analysis and Results: The manuscript talks about using a confusion matrix for statistical analysis, but it would be good to explain briefly how the matrix was used (for example, if any measures like accuracy, sensitivity, or specificity were calculated) and if statistical tests were done to check for significance. It would be better to add more details about the data analysis (for example, how the confusion matrix was used to calculate error rates) to make the analysis clearer. This will help the readers understand the validity of the results better.

Participant Sample Size: The sample size of 11 participants is small, and the manuscript should talk about the possible limitations because of this small number. It might also be helpful to mention the risk of low statistical power in finding meaningful effects with such a small sample. In the future, it could be good to include more participants to make the results more valid for a larger group.

Ethical Considerations: Although the manuscript says that participants gave informed consent, it would be better to explain if any other ethical matters were considered (for example, participant compensation, debriefing). It is also important to make sure that the informed consent form clearly explains the thermal stimulation procedure to the participants, because this might be uncomfortable for some people.
